# Priorities for Pan-American Geography Education: Needs and Trends

Alex Oberle [1,*] , Fabian Araya [2] and Sandra Alvarez [3]

1   Department of Geography, University of Northern Iowa, Cedar Falls, IA 50614, USA
2   Department of Social Sciences, University of La Serena, La Serena 599, Chile; faraya@userena.cl
3   Department of Education, University of La Serena, La Serena 599, Chile; salvarez@userena.cl
*   Correspondence: alex.oberle@uni.edu

**Abstract:** Geography education research in Latin America is uneven in terms of its thematic focus and country-level contributions. Research demonstrates the necessity for a region-wide prioritization of geography education themes that builds on existing scholarship and adapts to emerging needs and trends. The Pan-American Institute for Geography and History, a scholarly society, established a research team with a four-year charge of advancing geography education in the region. Titled "Geographic Literacy for the Countries of the Americas", the initiative develops research agendas that encompass the diversity of definitions and criteria, scholarly activities, and curricular resources across countries. The purpose of this study is to follow the process of identifying geography education research priorities involving scholars representing most countries in the region; describe the priorities and activities identified by project team members; and discuss emerging long-term research agendas. The study uses a descriptive research design to describe the process in general and to specify the outcomes. Results show the prioritization of the four areas as follows: teaching geography online, teaching geography face-to-face, methodological foundations of geographic education, and connecting geography and education stakeholders. Activities and research agendas within these categories include both traditional, longstanding themes and emerging themes related to recent global crises and technological innovations.

**Keywords:** geography education; Latin America; research agendas

## 1. Introduction

Geography education endeavors to bridge the two disparate fields of academic geography and research in education. Both disciplines are particularly fast changing, marked by recent shifts in the educational landscape and the evolution of geography driven by the increasing integration of geospatial technologies. The result is varying educational priorities depending on the region or country. Across Europe, particularly in the UK, geography education is grounded in the GeoCapabilities approach, a construct that combines robust disciplinary knowledge with individual empowerment [1,2]. Geography education in the United States is more diffuse, in part due to decisions about the curriculum occurring primarily at the state rather than federal level. Scholars have recently and more assertively sought to identify national-level priorities for geography education to include better support for elementary educators, integration of geospatial technologies into the classroom, resources for AP Human Geography, and professional development for pre-service teachers [3]. Similarly, researchers surveyed early career geography education scholars in the United States to determine their perspectives on priorities for geography education in the country and these include the need to enhance students' geographic skills, connect the work of professional geographers with elementary and secondary teachers and students, and promote the importance of the discipline across educational institutions and grade levels [4].

In Latin America, geography education scholarship is similarly diffuse and uneven in terms of the contributions from particular countries and the lack of alignment of disparate themes and priorities. This paper documents an ongoing systematic region-wide effort to advance geography education in the region, addressing the related questions of how a team of scholars identified geography education priorities across a large region with diverse national contexts and then in what ways are these priorities advanced through various initiatives and activities. The article begins with a review of the literature that chronicles both the evolution of geography education research and the larger priorities of the umbrella organization sponsoring this multi-year project. The essence of the paper then describes the project, focusing on its larger goals and the activities, to date, of the various working groups that drive the effort. Concluding the article is a discussion that outlines future plans for geography education in the region.

## 2. The Pan-American Institute for Geography and History

As the flagship organization in Latin America for advancing geography and history regionwide, the *Pan-American Institute for Geography and History* (commonly known by its Spanish acronym, IPGH) encompasses a mission that includes initiating and disseminating research, collaborating across commissions within the organization, and promoting interdisciplinary cooperation among other institutions and international organizations in the region [5]. The organization comprises all Spanish-speaking countries in Latin America as well as Brazil, Haiti, Belize, and the United States. To carry out its larger mission, IPGH operates the four commissions as follows: history, geography, geoscience, and cartography. An appointed board leads each commission and each hosts its own conferences. Commissions publish their own peer-reviewed journals as well as monographs that are often a collaboration with other commissions or organizations. The IPGH commission on geography furthers its goals through three committees on research, education, and communication dissemination. Each of these committees identifies their priorities based on the needs of the organization and its representative members. For example, a current priority for the committee on research is generating scholarship on climate change in the Americas.

Recognizing the need to elevate geography education in Latin America, where research has been limited by an unevenness in thematic focus and country-level contributions, the committee on education launched a four-year initiative called "Geographic Literacy for the Countries of the Americas". Following the country-level representation evident in the larger IPGH, the first step in this endeavor consisted of inviting geography education scholars from across the region. Ultimately, the effort includes nearly fifty researchers representing fifteen different countries. While the organization uses "Pan-American" as a regional construct, demonstrating its commitment to uniting the countries of South America, Central America, and North America, the locus of geography education contributions and its impact are from Latin American countries since the United States has only one representative on the project team and most of its geography education researchers are directly represented by the National Conference for Geographic Education (NCGE) since that US-based organization strives to navigate the complexities of the education landscape in a country where more policies occur at the state level rather than at the federal level. Canada, formerly an IPGH member years ago and potentially a member again in the future, is not currently part of the larger organization and is not part of the geographic literacy project. Since there is a membership fee and a process to follow for countries, or organizations within those countries, to join the IPGH, some nations have never elected to join or, like Canada, joined at one time but chose to no longer be a member.

## 3. Literature Review: Contextualizing Geography Education in the Americas

Geography education is critically important and relevant not just to careers and civic life but also fundamental to a well-rounded education and is necessary across all grade levels. As a sub-discipline that is conceptual, thematic, and pedagogical, geography education is at a critical juncture and in need of a reassessment of its conceptual core and

active lines of research. Central to this is a reevaluation of whether teaching and learning in geography is carried out in a manner that is consistent with both the contemporary conceptual and technological advances evident in the discipline at large. Over the past thirty years, scholars have contributed to the body of knowledge in geography education. In Europe, Rellou and Lambrinos enlisted geography education colleagues to undertake a country-by-country assessment of geography education, leading to recommendations to strengthen the field across the region [6]. These recommendations include establishing a common core or framework in geography education in all countries, identifying foundational elements of geography that transcend country boundaries and national curricula, and chartering a committee to promote region-wide European geography standards [6]. In addition, there are several contributions from Spain [7–15] and the English-speaking world through dedicated journals like *Journal of Geography*, *International Research in Geographical and Environmental Education*, and *Research in Geographic Education*. This scholarship, together with research from Latin America, established theoretical foundations and lines of research that respond to the needs and opportunities that exist in the educational context of contemporary Latin America.

Historically, by the end of the 1980s, there were fairly uniform approaches and perspectives on geography education across Latin America. Standard geography training for teacher education students consisted of a course of study embedded within the general pedagogical curriculum, sometimes as part of more specialized teaching method classes. Instructors for geography teaching methods courses were commonly geographers but geographers trained in a specific subdiscipline rather than the broader field, for example, in urban geography, social geography, or rural development. In other cases, the opposite occurred where instructors were not trained in the social sciences at all, hailing from colleges of education. Textbooks for geography teaching methods across various Latin American countries all originated from the same set of core scholarship. Harold Wood's *Course for Teaching Geography* (1980) commonly served as a foundational text as Wood was the president of the IPGH at that time [16]. The *Manual of Teaching Materials for Teaching Geography at the Secondary Level* (1983) was a highly relevant and influential text published by the IPGH in various Latin American countries [17].

Despite these contributions, geography education at both the elementary and secondary level at the end of the 1980s continued to be mainly descriptive and focused more on teaching methods and resources than on conceptual themes. Each of the elements and dimensions of geography and geographic space were studied independently and without an integrated vision. Geography education treated topics such as population and hydrology in a disjointed manner without any larger comprehensive geographic context. Cartography often became synonymous with geography and activities like drawing maps were overused in classrooms. In a similarly limiting manner, geographic themes, epistemologies, and teaching methods did not demonstrate any real relationship or connections with each other.

Geography education scholarship in Latin America began to transform in the 1990s and onwards. Drawing on various geography education publications, presentations, and conferences in the region over the last decade of the previous century, several trends emerged. First, the growing education reform process during this time demanded more effective and more specific teaching methods, including those used in geography. Overcrowded classrooms, a renewed emphasis on student learning, and the rise of educational theories like multiple intelligences, constructivism, critical pedagogy, and disciplinary performance standards energized educational systems and gradually established a new context for novel teaching methods and methodologies. Second, the 1990s marked a new era of teaching resources for geography education with new materials generated for students across the educational spectrum to include elementary, secondary, and post-secondary learners. These included texts with full-color photographs, cartographic applications for students, atlases, virtual resource centers, and new software. Slowly over time, educators adopted these new materials through the support of publishers, universities, ministries of education, formal and informal networks, and the internet. Concurrently, those developing

novel teaching materials then included a modernized perspective on geography education, shifting from the notion of geography being just an offshoot of history or an extension of natural science to one where it is its own discipline.

New information and communication technologies prompted the formation of academic and educational networks in geography education. Through these networks, teachers and researchers begin to recognize themselves as part of an academic community that shares common interests and challenges. Initially, members networked through personal contacts between academics who attended conferences, seminars, and workshops. This nascent network formation process began to achieve collaboration across Latin America between both teachers and researchers in geography education. Beginning in 2007, an academic online network enhanced scholarly work in geography education, but it neither did so across all of Latin America nor throughout the entire Pan-American realm. Comprising geography education researchers in Argentina, Brazil, Colombia, and Chile, the *Latin American Network for Teaching* Geography, or REDLADGEO using its Spanish acronym, has represented a foundational initial effort for regional collaboration that has eventually gone on to both host conferences every two years and continue the online forum for collaboration. The network publishes occasional reports and texts, especially in a Colombian national context, through the active research of the Society of Colombian Geography.

With a focus on geography education in Brazil, Colombia, and Chile, *Geographical Reasoning and Learning: Perspectives on Curriculum and Cartography from South America* represents a significant contribution to geography education in the region [18]. Scholars, through this edited volume, analyze learning with maps as well as fundamental topics such as spatial thinking, civic engagement, scientific literacy, and geography teaching. The book begins with a discussion of the nature and context of geography education in the three countries, providing a transferable framework for future efforts in the region that expand to all countries.

Cascante-Campos (2021) analyzes recent geography education research in Latin America, evaluating twenty years of open access journal publishing in the region from 2010 to 2020 [19]. The study documents consistent growth in geography education research but also an unevenness in terms of journal outlets and country-specific contributions. For example, several countries are not represented in the body of the literature while others, such as Brazil, are widely represented, with Brazil alone accounting for nearly 70% of the published peer reviewed articles in geography education [19]. Contemporary geography education themes in Latin America include five core categories, with theory, philosophy, and debates being the most published topical areas. Teaching methodologies and teacher preparation are also widely published, with instructional materials and resources rounding out the top five but with a more intermittent publication pattern [19].

Tracing how the priorities and strategies in geography education in the region can track with the flagship geography and history organization is important as it shapes the current project. Solis and Wintermute (2021) chronicle the progress of the IPGH during much of the 2010s up to 2021, documenting changing priorities and strategies that include expanded partnerships and enhanced capacity. The authors identify three strategic themes that ultimately support the efforts of the education in committee in its multi-year geography education endeavor [20]. Following the tradition of participation and inclusion in the larger IPGH organization, the authors cite the continuing effects of the global pandemic as one of their three questions to consider as the organization moves forward. Regarding technological changes, both those that are geography specific and those that are more broad-based, the article specifically cites artificial intelligence (AI) and volunteered geographic information (VGI) that is akin to citizen science. Evaluating the need for greater impact, the authors call for more external resources and partnerships to amplify the IPGH's efficacy [20].

## 4. Methods

This research employs a descriptive research design to evaluate, describe, and categorize the generalized process, goals/objectives, and outcomes of the "Geographic Literacy for the Countries of the Americas" effort to advance geographic literacy in the Americas [21]. With the multi-year project divided into work groups, each with a lead organizer who regularly reports to the larger project team, the research team can track the activities of each work group to follow the process and describe the outcomes. These work team reports consist of discussions, presentations, written notes and narratives, and webinars. From the outset of the project and typically once a month, the project team convenes whole group meetings to maintain communication and continuity and for establishing and revising goals and strategies.

## 5. The IPGH Committee on Geographic Education

Geography education comprises a discipline that has great potential for training present and future citizens of their respective countries and the larger global community. One of the most important purposes of geography education is to contribute to the development of geographic thinking as both a worthy disciplinary and pedagogical pursuit. Considering contemporary problems from all spatial scales, the discipline must promote the development of a human–environment relationship that is based on an integrated perspective of geographical space. The IPGH's education committee endeavors to work from 2022 to 2025 to advance "Geographic Literacy for the Countries of the Americas". Through this project, the leadership team has established work groups with representatives from each of the IPGH member countries. Each work group's plan allows for the defining of the goals/objectives, activities, and curriculum resources that support the broader endeavor. This, along with knowledge dissemination, will form the priorities and pathways for advancing geographic literacy in the Pan-American context with a focus on Latin America.

Through this project, the research team expects to reduce existing inequalities in terms of access to geographical knowledge across the countries of the Americas. To accomplish this, the project team will invite other stakeholders who are attuned to geography education and enthusiastic about addressing issues in the field combined with online networks and other internet-based avenues facilitating collaboration. This project aims to develop specific lines of research and teaching called "teaching methods for the development of geographic thinking". Through this, the project team will research the links between teaching methods, student learning, and understanding spatial relationships and interdependencies.

The role of geographic education in developing geographic thinking skills is uniquely relevant, preparing students to take action in a world where there is a great deal of local and global mobility and where such skills allow them to interact with and understand the dynamic relationships that comprise natural and human systems. Thus, the skills central to geographic thinking are essential for a range of everyday actions like negotiating places, both known and unknown, playing sports or games that involve spatial thinking related strategies, and effectively analyzing information in maps, charts, and diagrams. The IPGH geographic education commission and its members assert that the most effective way to make decisions and take informed action in an increasingly globalized, complex, and interdependent world is through fostering geographically informed citizens. Education, in the broadest sense of the concept, is the only means for achieving the type of social and environmental transformations needed to benefit individual countries and the larger region.

## 6. Project Guidelines and Work Plan

The project includes three types of goals of strategic, academic, and professional development (Table 1). Strategic goals are those ideally designed to build capacity as the project advances, such as a goal of leveraging the existing IPHG YouTube channel to establish a dedicated pathway for online dissemination, including a short video series called "Panamerican Geo-vizualization" that will host livestreamed and recorded geography education

presentations and roundtable discussions. Academic goals encompass action steps that primarily reach a higher education audience of professors, researchers, and teacher education programs. These include the signature academic goal of establishing a mechanism for professional and technical assistance related to geographic literacy, integrating geographic thinking and geographic inquiry around core issues in the region such as climate change, nature–society themes, and civic engagement. Four strategies comprise the professional development goal category, including expanding online networks to reach educators and organizing both in-person and online events to address fundamental or timely topics in Pan-American geography education.

**Table 1.** "Geographic Literacy for the Countries of the Americas" project goals.

| Strategic Goals |
| --- |
| Establish the online "*Pan-american Geo-visualization*" series to host presentations and roundtable discussions |
| Create online manuals and digital texts to disseminate the relevance of geography education |
| Coordinate an online professional network of Pan-american geography education scholars and stakeholders |
| **Academic Goals** |
| Prepare a document that notes challenges in geography education in the region |
| Draft a document that amplifies existing work in geographic inquiry towards a diverse audience |
| Establish a mechanism for providing professional and technological assistance related to geographic literacy |
| **Professional Development Goals** |
| Expand online networks and both online and in-person events that reach classroom educators |
| Build and curate a repository for disseminating best practices that serve the whole region |
| Invite teachers and students to participate in the new online networks and special events |

## 7. Committee for Geography Education Working Groups: Descriptions and Challenges

"Geographic Literacy for the Countries of the Americas" comprises representatives from each member country, forming working groups that are dedicated to achieving the larger project, and working group specific objectives. These individual participants also represent their respective countries and are expected to work on behalf of their national context, structures, and stakeholders, knowing their country's educational landscape, and reaching out to colleagues or institutions in that country for further input. The project team as a whole identified core requirements for geography education and the working groups then coalesced around these to operationalize both the objectives and strategies. This is well demonstrated through a primary initial activity of the working group of "connecting geography and education stakeholders" and the creation of a survey on geography education at the pre-school and elementary levels. Table 2 highlights the questions in this survey that was administered to one project team representative from each country. Dividing the questions so that respondents can answer each for pre-school and elementary education categories, the survey encompasses disciplinary, curricular, pedagogical, and professional preparation categories. The survey results are not only foundational for this working group's charge but are also utilized by the project team as a whole to develop or refine strategies and goals/objectives.

**Table 2.** Survey questions to determine pre-school and elementary geography education needs and priorities.

| Disciplinary Scope |
| --- |
| The number of years of geography/geographic themes assigned |
| The number of hours per week that geography is taught |
| **Geography Curriculum** |
| The types of geography curriculum (e.g., stand-alone, integrated with social studies, integrated with science) |
| Number of courses that include geography themes |
| The names of courses that include geography themes |
| The number and type of learning objectives that include geography themes |
| The type of geography content (e.g., fundamentals of geography, physical geography, population geography, economic geography, political geography, local/regional/national geography, world geography) |
| The universal themes and content that support geography (e.g., civic engagement, technology, environmental education, inquiry skills, respect for cultural diversity, etc.) |
| **Pedagogical Scope** |
| The types of teaching materials and resources used in geography classes (e.g., globes, Google Earth, atlases, books and magazines, audiovisual material, etc.) |
| The predominant teaching methods and strategies used in geography classes (e.g., inquiry, collaborative work, cartographic analysis, fieldwork, observation, etc.) |
| The ways that geography teaching is evaluated (e.g., geographic skills, understanding of geographic systems and space, values and responsibilities related to geography, etc.) |
| **Professional Scope: Teacher Preparation** |
| The number of years of teacher education training that a typical teacher receives |
| The type of teacher education training that a typical teacher receives (e.g., technical institution, normal school, university, etc.) as well as whether it is a private or public institution |
| The existence of any required entrance exam to enter the profession |
| The origin of guidelines/accreditation for teacher education (e.g., national, state/region, etc.) |

The working group subgroups additionally serve to disseminate the expanding knowledge base in their respective areas, institutions, and countries, employing different modalities and pathways for sharing their expertise. Operating with a great deal of autonomy, each working group designed a roadmap to demonstrate how their themes, objectives, and activities contribute to the general goals of the committee on the geography education project. As expected for a multi-year project, the four working groups are at varying stages, some only at the initial stages of their plan while others, chiefly the "teaching geography online" working group, have achieved the core goals and completed activities that reach a wider audience.

Globally, regardless of the region, geography education tends to be split between the discipline of geography and often separate academic structures such as colleges of education or teacher preparation programs. As such, it is critical to include both traditions and foundations in geography education and that is the reason for the establishment of the working group called "connecting geography and education stakeholders". The primary goal of this team is to link the evolving work of the whole IPGH project with the various communities and stakeholders in the Pan-American education system. In the education environment of this region, the primary driver of educational quality is an effective partnership between educational institutions and external organizations.

The "geography teaching methods" working group endeavors to provide better guidance on teaching methodologies to establish more effective processes and practices for

teaching and learning in geography across all IPGH member nations. This need has become more urgent and relevant because of the growing value placed on teaching methods by ministries of education and school administrators. The working group intends to launch webinars as a primary means to disseminate the best practices that they are curating. A primary challenge, of course, is the myriad country- or even school-level differences that make a broader multi-country regional approach difficult, especially as the aim of this project is broader dissemination and wider applicability and not a focus that only serves a few countries or types of schools.

Two of the four working groups seek to advance geography teaching, separating it into a face-to-face teaching working group and an online teaching working group. The "teaching geography in person" working group endeavors to highlight the development of skills and related perspectives that are critical in the face-to-face teaching environment, with a special focus on field trips and field-based projects. To date, the group is collecting information from each of the IPGH member countries regarding best practices with the goal of creating a virtual repository where educators can access and view lesson plans, activity descriptions, and both photographs and videos of exemplary classroom practices.

*Working Group: Teaching Geography On-Line*

With the rapid advent of new educational technologies, such as robust learning management systems and discipline specific tools, contemporary teaching and learning increasingly leverage educational technologies to further students' abilities in constructing, collaborating, and creating [22]. The effects of the COVID-19 pandemic required educators to quickly employ these technologies to develop on-line teaching programs. The combination of technological advances and the varied outcomes of on-line teaching during the pandemic prompted the larger project team to identify on-line teaching as one of the priority themes for "Geographic Literacy for the Countries of the Americas". As such, a working group emerged around this theme to develop a plan and series of activities to enhance on-line learning in geographic education. The working group consists of seven geography education scholars from six countries who meet regularly to determine priorities within the larger theme of geography education. The group identified the priorities based on the needs of the geography education community in their own country, the group members' expertise, and existing opportunities to discuss and disseminate these priorities.

This working group is far along in their activities, including already hosting an on-line panel series and issuing a call for papers for a special edition of a publication. Together, these activities further the goal of the larger project and improve teaching strategies, promote the geography curriculum, and highlight the importance of geographic literacy. The panel series addresses emerging debates in geography education related to technology. The theme for the first panel is artificial intelligence (AI), a pressing topic across education in all disciplines but one presenting unique challenges and opportunities in geography education. Following the focus on emerging and timely topics, the second panel discusses "fake news", similarly a concern across disciplines but especially pertinent to geography education. For example, with the prioritization of climate change research in geography, disinformation abounds with falsehoods that seek to minimize or discredit research in climate science. Many aspects of human geography touch on geopolitics, another area where propaganda seeks to distort the facts. Concurrent with the panel series, the work group secured a commitment from a Brazilian geography journal, *Ateliê Geográfico*, to publish a special issue on the Pan-American context for geography education.

## 8. Discussion

"Geographic Literacy for the Countries of the Americas" endeavors to serve as a catalyst for continued efforts to systematically advance future ideas, initiatives, and projects in geography education that will start in 2025 when the formal IPGH multi-year project concludes. The various aspirational goals and strategies presented here begin with those that are more general and lead to those that are more focused, responding to the larger need

for developing geographic thinking as a fundamental element of geography education. As this effort arrives at its midpoint, it is important to assess the progress so far. Table 3. outlines the planned 2024 signature activities for each of the working groups, the second to last year of the project (2025, the final year of the project, consists of intensive dissemination rather than any new or expanded initiatives). Overall, the project is making solid progress, with two of the four subgroups clearly on track for their 2024 activities. This includes the "teaching geography in-person" working group that endeavors to organize an in-person conference in Brazil. The two on-line webinars in 2023, focusing on AI and "fake news" in terms of subject matters that can be included, serve as a foundation for an in-person conference, and the leading organization is a Brazilian university that facilitates logistics. The "connecting geography and education stakeholders" working group has had a head start in rolling out a revised version of a geographic inquiry process due to the leader for that group assisting in the piloting of a National Geographic-developed model among a teacher education program and regional schools in Chile. The other two subgroups have ample time to make additional progress in the next year to realize the goals and activities they established in their project contribution timeline.

**Table 3.** Planned working group signature activities for 2024.

| Working Group | Activities to Be Completed for 2024 |
| --- | --- |
| Working group: teaching geography on-line | Develop an online course on geographic thinking in a Pan-American regional context |
| Working group: teaching geography in-person | Organize an in-person conference or seminar in Brazil that focuses on specific themes in geography education in the region |
| Working group: geography teaching methods | Expand the resource base of digital texts and videos that promotes the relevance of geography teaching in the region |
| Working group: connecting geography and education stakeholders | Establish geographic inquiry curricula and projects in select countries |

### 8.1. Global Scale

At a global scale, beyond IPGH membership in Latin America and North America, the next step for this project seeks to generate greater collaboration around the common mission, goals, and strategies of international geography education organizations including the International Geographical Union (IGU) that has strong representation in Europe, Asia, Australia, and other countries; the European Association of Geographers (EUROGEO); and Spain's Geoforo. In each case, geography education scholars who contributed to the IPGH project will systematically review documents, conference summaries, websites, webinars, and publications for each of the three organizations to identify strategies and best practices for further developing geographic thinking in IPGH member countries as well as capitalizing on opportunities to contribute to complementary global initiatives outside the region and across the globe.

### 8.2. Pan-American Scale

Leveraging longstanding and emerging collaborations within IPGH member countries as well as developing potential new partnerships in the Americas will ensure that "Geographic Literacy for the Countries of the Americas" builds on its accomplishments and continues beyond the lifetime of direct IPGH support. The project team leaders have established research agreements with other universities including a university in the United States with a core nucleus of geography education researchers. Through this agreement, IPGH geography education scholars will draft a research agenda that includes new geographic literacy initiatives. The National Council for Geographic Education (NCGE) is the flagship academic geography education organization in the United States and one that

is interlinked with the IPGH since the US is an active member country. The Canadian Association of Geographers (CAG) is the academic geography organization for Canada and includes geography education scholars. IPGH education scholars will capitalize on the completion of larger projects in 2025 to review NCGE and CAG priorities, determining concrete avenues for mutually supportive collaboration. Since geography education is a committee within the geography commission of the IPGH and geography is one of four commissions in the larger IPGH organization, the continuation of active collaboration and advocacy within the IPGH is critical. Existing opportunities for teamwork abound in the organization including grant funding that encourages collaboration across multiple commissions as well as the priority areas identified by the commission on geography, such as climate change, gender, spatial analysis, and hazards.

## 9. Conclusions

The IPGH committee on geography education, through its "Geographic Literacy for the Countries of the Americas" initiative, achieves a critical contribution to the field of geography education that transcends the Americas and elevates the discipline's global impact. The effort enlisted a research team with members from each IPGH country who represented geography education in their nation to establish a set of priorities for the region as a whole that were also transferable to the contexts of individual countries. Once divided into working groups, the project team could then advance specific strategies and goals while still heeding the objectives and priorities of the initiative. By modernizing geography education in the region, aligning priorities to address past disparities among country and thematic contributions, and bridging academic geography with stakeholders and research in education, the endeavor capitalizes on a range of opportunities and tackles myriad challenges. At the regional level, the project creates curricular materials and content as well as widely accessible best practices and knowledge bases that are designed to be readily applicable to multiple countries. Globally, the endeavor contributes to wider scholarship on on-line learning and understanding pathways for connecting the work of academic geographers with practices in the field of education.

There is high potential for continuing future collaboration that leverages the priorities of the IPGH with new or strengthened partnerships within the Americas as well as collaboration with other regional and international geography education organizations. With the continued expansion of partnerships, the IPGH can link any emerging geography education initiatives with governmental organizations, non-profits, and corporate entities whose mission and strategies align with education and teaching. Similarly, the organization's aspiration to further volunteered geographic information and citizen science can support ongoing activities in geography education that relates both to field trips/fieldwork and on-line teaching and learning. Future collaboration with organizations like the IGU, NCGE, and CAG can align priorities in those programs with those that serve IPGH member countries as well as, potentially, GeoCapabilities, utilizing geospatial technologies in primary and secondary education and pathways that connect with teacher education students.

**Author Contributions:** Conceptualization, F.A., S.A. and A.O.; methodology, A.O., F.A. and S.A.; investigation, S.A., F.A. and A.O.; writing—original draft preparation, A.O., F.A. and S.A.; writing—review and editing, A.O., F.A. and S.A.; project administration—F.A. and S.A. All authors have read and agreed to the published version of the manuscript.

**Funding:** This research received no external funding.

**Institutional Review Board Statement:** There is a documentation from the Associate Director of the Office of Research and Sponsored Programs at the University of Northern Iowa regarding IRB/Human Subjects, confirming that this research does not constitute human subjects research.

**Informed Consent Statement:** Not applicable.

**Data Availability Statement:** Dataset available on request from authors.

**Conflicts of Interest:** The authors declare no conflict of interest.

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
