# Peer review of "Priorities for Pan-American Geography Education: Needs and Trends"

_education, doi:10.3390/educsci14010064_

Round 1
Reviewer 1 Report
Comments and Suggestions for Authors
This is a well written article with information on a wonderful project that could be beneficial for the Geography education community. However, for that information to be made useful, it needs to be better compiled and organized, e.g. the use of tables, sharing of data that is referred to in the text (identification of specific priorities, goals, strategies).
The Method section promises evaluation, description and categorization of process and outcomes, but there was no indication of evaluation criteria or process for evaluation, nor did I find categorization of working group goals or outcomes, which would actually be very helpful information. Keeping a lens on these goals, throughout the article, could be better achieved through a solid research question addressed by the descriptive research methods and referred to again in the Discussion and Conclusions.
Having an evaluation of the process of coordinating a regional initiative like this, even a baseline that could be compared in future years (as there doesn't yet seem to be much to report from three of the groups), and lessons learned along the way would be a strong contribution to the literature, and provide measurements of success for the Geoliteracy project as it unfolds.
Specific comments are provided in the attached file.

Author Response
Dear Reviewer:
Please see the attached .pdf document regarding our response to your recommendations.

Reviewer 2 Report
Comments and Suggestions for Authors
This is a work that provides good information about what is planned to be done in Latin America for the advancement of geographic education. The authors present extensively the Institute and its activity as well as the relationships it has and others it will develop for the promotion of geographic education. However, they do not provide information about the current state of the school geography curricula in the primary and secondary education level of Latin American countries. The reader does not know if there have been any studies that establish the differences and similarities that exist in geography education between the countries in order to be able to justify what is presented in the article. I guess some studies have been done but to what depth? how many countries' curricula have been studied and compared?
I think such information would help the reader understand how common the problems mentioned are and whether efforts have been made to approach them in school geography.
I therefore expected the work to be enriched with some such references and comments on these.
To help I could recommend a publication that does just that for European countries.
"Rellou, M. and Lambrinos, N., 2008. The school geography curriculum in European Geography education: Similarities and differences in the United Europe. In: "European Geography Education: The Challenges of a New Era", Nikos Lambrinos and Maria Rellou (eds.), Pathways in Geography Series No. 36, National Council for Geographic Education, Washington D.C., pp. 1-20."

Author Response

(The authors gave the same response as above.)
